# Structural and functional characterization of the bacterial biofilm activator RemA

Tamara Hoffmann[1,5], Devid Mrusek[2,5], Patricia Bedrunka[2,5], Fabiana Burchert[2], Christopher-Nils Mais[2], Daniel B. Kearns [3], Florian Altegoer [2], Erhard Bremer [1✉] & Gert Bange [2,4✉]

*Bacillus subtilis* can form structurally complex biofilms on solid or liquid surfaces, which requires expression of genes for matrix production. The transcription of these genes is activated by regulatory protein RemA, which binds to poorly conserved, repetitive DNA regions but lacks obvious DNA-binding motifs or domains. Here, we present the structure of the RemA homologue from *Geobacillus thermodenitrificans*, showing a unique octameric ring with the potential to form a 16-meric superstructure. These results, together with further biochemical and in vivo characterization of *B. subtilis* RemA, suggests that the protein can wrap DNA around its ring-like structure through a LytTR-related domain.

[1] Philipps-University Marburg, Center for Synthetic Microbiology (SYNMIKRO) & Faculty of Biology, Marburg, Germany. [2] Philipps-University Marburg, Center for Synthetic Microbiology (SYNMIKRO) & Faculty of Chemistry, Marburg, Germany. [3] Department of Biology, Indiana University, Bloomington, IN, USA. [4] Max Planck Institute for Terrestrial Microbiology, Marburg, Germany. [5]These authors contributed equally: Tamara Hoffmann, Devid Mrusek, Patricia Bedrunka. ✉email: bremer@staff.uni-marburg.de; gert.bange@synmikro.uni-marburg.de

Many bacterial species can form biofilms, assemblages in which cells of a single or of multiple species co-exist in a self-produced extracellular polymeric matrix[1,2]. Biofilms are not only important from an eco-physiological point of view, but they also provide considerable medical and environmental challenges[3], while at the same time also offering new opportunities in biotechnology and bioremediation[4,5].

One of the microorganisms in which the genetics and physiology of biofilm formation have been intensively studied is the Gram-positive soil bacterium *Bacillus subtilis*[2,6–8]. *B. subtilis* can form biofilms on solid surfaces (macro-colonies) and at liquid/air interfaces (pellicles)[9]. These traits enable *B. subtilis* to colonize plant roots, thereby allowing the cells to take advantage of nutrients present in root exudates[10–13]. In order to colonize root surfaces, *B. subtilis* cells need to switch from the planktonic motile to the sessile non-motile lifestyle in which the cells are embedded into a genetically and physiologically heterogeneous, and structurally complex community[2,6,7,14]. This microbial assemblage is encased in an extracellular polysaccharide matrix (EPS), which also contains the extracellular TasA protein[15,16]. Macro-colonies of *B. subtilis* growing on solid surfaces are covered by the redox-sensitive hydrophobin BslA, thereby providing the biofilm with water-repellent attributes[17]. The environmental and cellular cues that trigger the switch from a motile to a sessile lifestyle of *B. subtilis* are not completely understood, but centrally involve the two membrane-embedded potassium responsive sensor-histidine kinases KinC and KinD[18,19]. Their sensory output is transferred through the phospho-relay[20] to the response regulator Spo0A, a master regulator of cellular differentiation in *B. subtilis* (summarized in:[6,8,14,21]).

For the onset and progression of biofilm formation, an intermediate threshold level of phosphorylated Spo0A (Spo0A-P) is needed[22]. Under these conditions, transcription of the *sinI* gene encoding an antagonist (SinI) of the SinR repressor protein, another important regulator of biofilm formation[23–25], is upregulated[26–28]. Resulting from the sequestration of the SinR protein by SinI into a protein complex unable to bind DNA, transcription of operons involved in matrix production (*epsA-O*), synthesis of the amyloid-like protein TasA (*tapA-sipW-tasA*), and the gene for the BslA hydrophobin are de-repressed, thereby fostering biofilm formation[6]. In this process, an epigenetic switch couples biofilm formation with inhibition of motility through SlrR, another antagonist of SinR, where the SinR:SlrR heterodimer serves as a DNA-binding complex to repress autolysin and motility genes[29]. De-repression of *epsA-O* expression also contributes to loss of motility because this operon encodes EpsE, a bi-functional protein not only involved in EPS biosynthesis, but which also serves as a clutch, thereby directly inhibiting flagellar rotation[30,31].

A suppressor screen aimed to identify mutations that restored motility to a *sinR* mutant discovered changes in two genes, referred to as *remA* and *remB* (regulator of extracellular matrix genes)[32]. While little is known about RemB, further studies focusing on RemA identified this protein as a transcriptional activator for all three major biofilm-promoting operons/genes[33]. Even in a *sinR* mutant where the activity of the *epsA-O*, *tapA-sipW-tasA*, *bslA,* and *slrR* promoters are de-repressed, a functional RemA protein is crucial for biofilm formation[32]. RemA is a protein with a molecular weight (MW) of 9.6 kDa and lacks recognizable DNA-binding features; yet it does specifically interact with DNA regions upstream of its target promoters[33]. At its binding regions, RemA does not leave footprints typical for canonical bacterial repressors and activator proteins. Instead, RemA exhibits multiple and closely spaced DNase I protection sites, which are AT-rich but otherwise only share a low degree of conservation[33]. An in-depth mechanistic understanding of this important regulator of *B. subtilis* cellular differentiation is lacking.

Here, we show that RemA organizes into a structurally unique 8-meric ring, which can further assemble into a 16-meric superstructure. Our biochemical and functional data suggest that RemA binds its target DNA through the outer surface of its ring-like topology through a DNA-binding domain related to LytTR. Taken together, our study enables us to reconcile previous data[33], and to propose a genetic model for the interaction of RemA with the SinR-controlled *epsA* and *tapA* regulatory regions.

## Results

**Biochemical analysis of recombinant RemA proteins suggests oligomer formation**. To gain a deeper understanding into the structural basis of the activating activity of the biofilm regulator RemA[32,33], we sought to determine its crystal structure. However, crystallization trials with the *B. subtilis* (*Bs*)RemA were unsuccessful. One major challenge was the recombinant production and solubility of (*Bs*)RemA in *Escherichia coli* either as untagged or hexa-histidine (His)-tagged protein, as also noted previously where (*Bs*)RemA could only be produced and kept soluble as an N-terminal maltose-binding protein (MBP) fusion protein[33].

As proteins from thermophilic organisms are often advantageous for biochemical and structural studies, we chose RemA from the moderate thermophile *Geobacillus thermodenitrificans* (*Gt*) instead. The (*Gt*) and (*Bs*)RemA proteins share 66% amino acid sequence identity and exhibit over 95% conservation (Fig. 1a). The (*Gt*)RemA protein containing an N-terminal His-tag was purified by a two-stepped protocol consisting of a Ni-ion affinity followed by a size exclusion chromatography (SEC) step. Multi-angle light scattering analysis (MALS) of the two major SEC peaks of (*Gt*)RemA (MW = 9.6 kDa; 87 amino acids) indicated two particles with the approximate molecular weights of 135 kDa ± 15% and 77 kDa ± 10%, which might correspond to 16-mers $(RemA)_{16}$ and 8-mers $(RemA)_8$ of the protein, respectively (Fig. 1b). The ratio of both peaks was dependent on the protein concentration, indicating that higher concentrations promote the formation of $(RemA)_{16}$ (Supplementary Figs. 1a, b). These findings suggest that RemA can form 8- and 16-mers in a concentration-dependent manner.

To determine whether (*Bs*)RemA would also form higher oligomers, we employed an N-terminal (His)-GB1-tagged fusion variant, which could be purified as a soluble protein (GB1: B1 domain of Streptococcal protein G[34]). Of note, removal of the (His)-GB1 tag through a (Tobacco Etch Virus nuclear-inclusion-a endopeptidase) TEV cleavage site present between the solubility tag and (*Bs*)RemA resulted in immediate protein precipitation, which was in stark contrast to the equivalent construct made with the (*Gt*)RemA. However, SEC analysis of both the (*Bs*)RemA and (*Gt*)RemA (His)-GB1 constructs suggested the exclusive presence of octamers (Supplementary Figs. 2a, b). Collectively, these findings suggest that solubility tags (such as MBP or GB1), which are much larger than a His-tag, affect the oligomerization properties of both RemA proteins (see discussion).

**Structural analysis of RemA reveals its ring-like architecture**. The structure of (*Gt*)RemA was determined at 2.3 Å resolution by selenium single-wavelength anomalous diffraction (Se-SAD), because no appropriate search model could be identified (Supplementary Table 1). Amino acids 2–83 of (*Gt*)RemA could be unambiguously assigned into the electron density map. The overall structure of a RemA monomer shows a novel, wedge-shaped domain with a β-β-α fold, consisting of six β-strands and two α-helices (Fig. 1c, PDB-ID: "7BM2 [https://doi.org/10.2210/pdb7BM2/pdb]"). Analysis of the crystallographic asymmetric unit showed that four RemA monomers arranged into a tetramer reminiscent of a semicircle. Following the crystallographic 2-fold

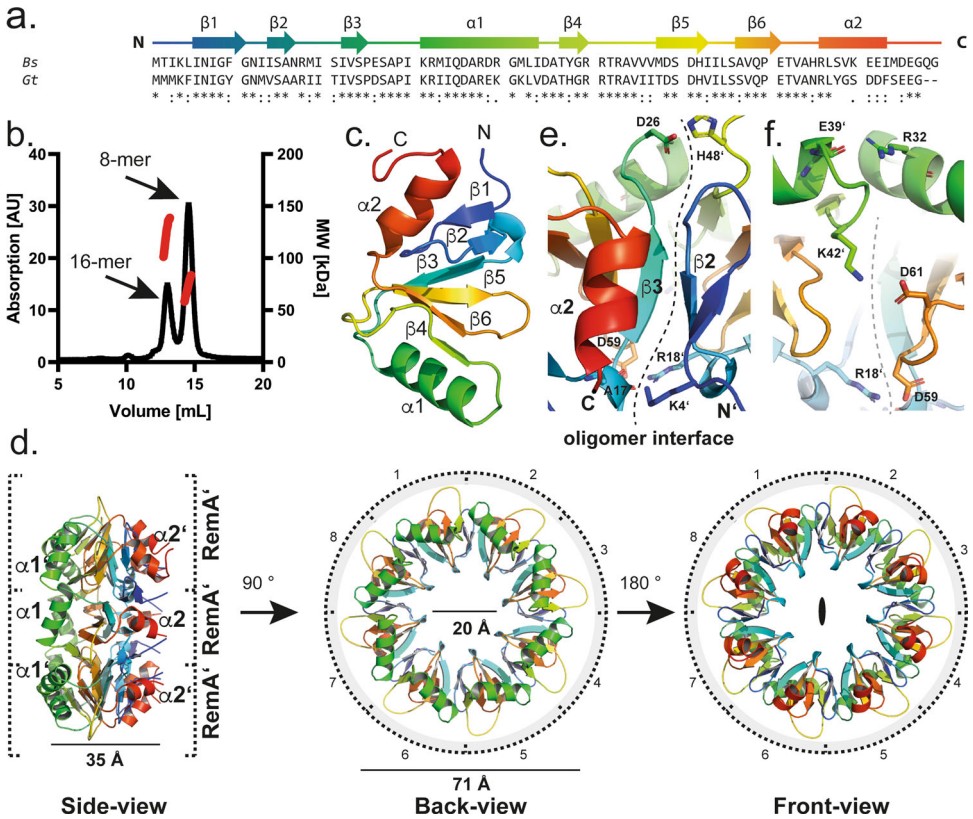

**Fig. 1 Structure of the biofilm regulator RemA. a** Amino acid sequence alignment of RemA from the Gram-positive model organism *B. subtilis* (*Bs*) and the moderate thermophilic bacterium *G. thermodenitrificans* (*Gt*). The secondary structural elements observed in the model are drawn schematically above the alignment and colored in rainbow colors from the N- to the C-terminus. **b** Chromatogram of the analytical SEC-MALS experiment of (*Gt*)RemA at the protein concentration of 500 µM. The absorption at a wavelength of 278 nm and the molecular weight (MW) calculated from MALS are in black and red, respectively. Source data are provided as a source data file. **c** Cartoon representation of a RemA monomer in rainbow colors from the N- to C-terminus (indicated by: 'N' and 'C', respectively; PDB-ID: "7BM2"). **d** Side, back, and front views of a (RemA)8. The color code is as in Fig. 1a, c. **e-f** Contact interface between two RemA subunits within (RemA)8. Further description is given in the main text.

rotation axis, the other half of the semicircle is found in the neighboring asymmetric unit across the crystallographic two-fold axis (Supplementary Fig. 3a). Thus, our structural analysis, consistent with our biochemical analysis, shows that 8 monomers of RemA form a donut-shaped 8-mer with outer dimensions of approximately 71 and 35 Å and a central hole of approximately 20 Å in diameter (Fig. 1d; left and middle panel). The buried surface area between two RemA monomers is 955 Å$^2$, summing up to an interface area of approximately 7640 Å$^2$ between all eight subunits within the RemA ring.

Within the (RemA)8, each of the monomers primarily interacts via hydrogen bonds and salt bridges with the two neighboring molecules. Central for the interaction of two monomers within the ring is the β-sheet augmentation of β2 of one monomer with β3 of the adjacent monomer (Fig. 1e). Several interactions, which localize on the outer and inner sides of the ring, foster the stabilization of the monomers within the ring and of the overall geometry of the 8-mer (Fig. 1f; see also below). The RemA structure also shows that the two helices α1 and α2 present in each of the monomers, decorate the back- and front sides of (RemA)8, respectively (Fig. 1d). Taken together, our structural analysis reveals that RemA forms highly symmetric superstructures possessing an overall donut-like shape.

**Multiple binding sites form a continuous DNA interaction interface.** Closer inspection of our RemA structure with the aim to identify its putative DNA-binding motif(s) or domain(s) revealed an extended, positively charged surface on the outside of the 8-mer (Fig. 2a). Moreover, this surface coordinates three

sulfate ions (i.e., S1 to S3) originating from the crystallization buffer. These sulfate ions are primarily coordinated by the positively charged arginines 50, 51, and 53 (Fig. 2b, c). Hence, the sulfate ions might mimic backbone phosphate groups of a DNA molecule as often observed in structures of nucleic acid-binding proteins (e.g.,[35]). Moreover, a structural homology search using the DALI server[36] revealed structural similarity of RemA to the LytTR-type DNA-binding domain found in the response regulator AgrA (Supplementary Fig. 3b), the global regulator of virulence in *Staphylococcus aureus*[37,38]. Two adjacent subunits of (RemA)8 superimpose well with the C-terminal binding domain of AgrA (AgrA-C, PDB-ID: "3BS1 [https://doi.org/10.2210/pdb3BS1/pdb]") with a root mean square deviation (r.m.s.d.) of ≈1.4 Å over 165 Cα-atoms (Fig. 2d). A closer inspection of this superimposition revealed that R170 and R218 at AgrA-C involved in coordinating the phosphate backbone of the DNA are near the arginines 50, 51, and 53 on the outer surface of the donut-like RemA8 structure (Supplementary Fig. 3e). This suggested the role of these arginines in the DNA-binding of RemA.

To challenge this idea, we performed electro-mobility shift assays (EMSAs) with wild type (*Gt*)RemA and variants in which the central arginines 50 and 51 were exchanged to alanines. While wildtype (*Gt*)RemA was able to shift DNA fragments containing the *epsA* promoter region (P*epsA*) (Fig. 2e), a known target of RemA[32,33], neither the (*Gt*)RemA-R50A nor the -R51A variant was able to shift DNA (Fig. 2e). The (His)6-tag present at the N-terminus of RemA did not affect the binding of the wildtype protein to DNA fragments containing P*epsA* in the EMSAs,

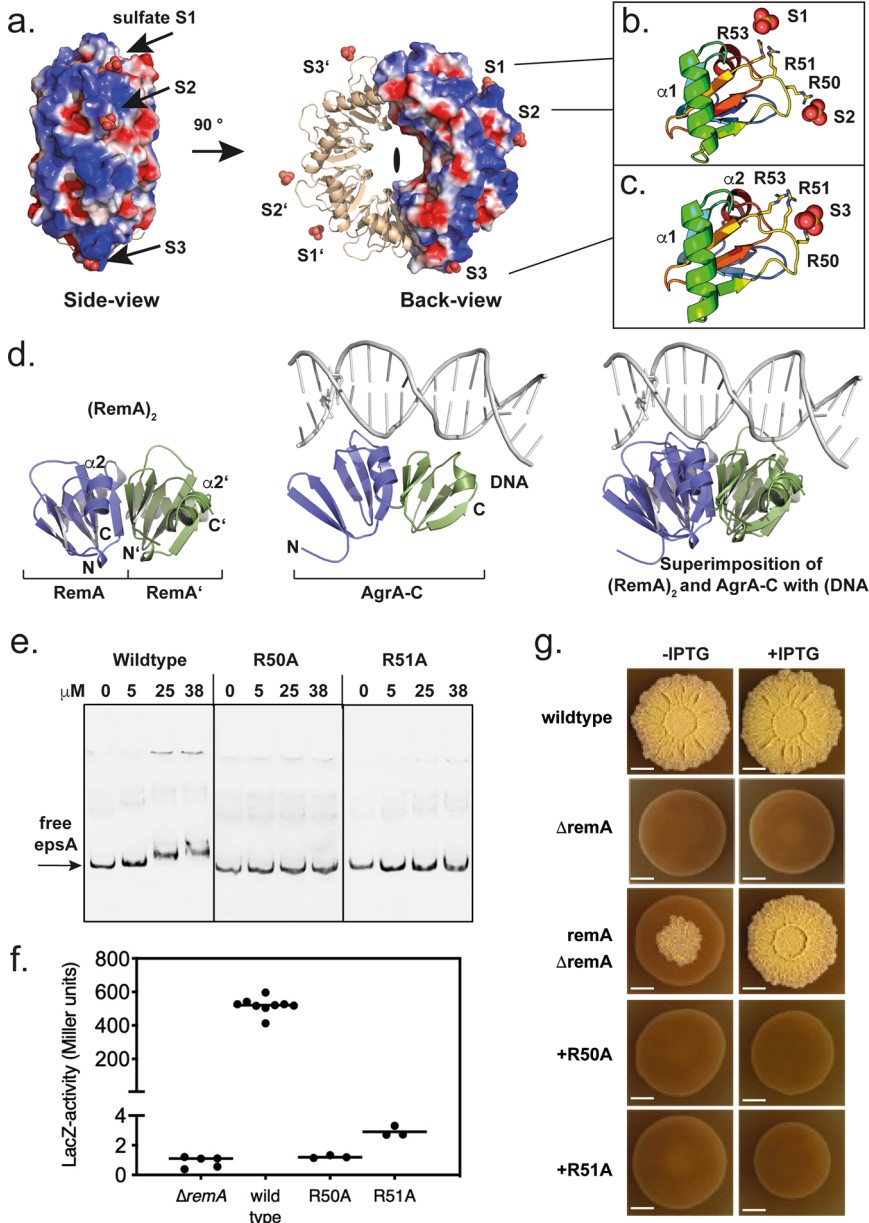

**Fig. 2 DNA-binding by RemA. a** Electrostatic surface of the RemA 8-mer (side and back view, PDB-ID: "7BM2"). 'S' indicates sulfates found at the surface of the crystal structure. A detailed view on the coordination of the sulfates S1, S2, and S3 is given in figure panels (**b**) and (**c**), respectively. **d** Structure of a RemA dimer of (RemA)$_8$ (left side; PDB-ID: "7BM2"), AgrA-C bound to DNA (middle; PDB-ID: "3BS1"), and the superimposition of both (right side). **e** EMSAs of wild type RemA, RemA-R50A, and RemA-R51A with a DNA fragment containing the regulatory *epsA* region. Results were confirmed with three independent preparations of recombinant proteins. **f** β-galactosidase activity in strains carrying P$_{epsA}$-*lacZ* fusions. Activity assays of a *remA* mutant strain ($n = 5$) and of complementation strains encoding inducible (+ IPTG), ectopic copies of wild type RemA ($n = 9$), RemA-R50A ($n = 3$), and RemA-R51A ($n = 3$). Each point reflects the LacZ activity measured in a biological replicate. Data are presented as mean from independent experiments. Source data are provided as a source data file. **g** Biofilm assays of wildtype *B. subtilis* 3610, a *remA* deletion strain (Δ*remA*) and its complementation with wildtype RemA (Δ*remA* + *remA*), *remA*-R50A (R50A), and *remA*-R51A (R51A). The length of the scale bars corresponds to 0.5 cm. Biofilm formation was analyzed in at least two independent experiments for each strain with several independently grown colonies per experiment. The colonies shown represent typical examples.

because a tag-free variant of (*Gt*)RemA showed the same DNA-binding behavior as its His-tagged counterpart (Supplementary Fig. 3f). These findings show that DNA-binding by RemA involves the arginines 50 and 51.

To investigate whether the variation of these residues impacts the structure of (RemA)$_8$, we aimed at crystallizing the (*Gt*) RemA-R50A and -R51A variants. Unfortunately, both variants did not crystallize, likely due to the lack of the arginines 50 and 51, which are relevant to establish crystal contacts involving the sulfate ions (Fig. 2a, b). However, a (*Gt*)RemA variant in which

R51 and R53A were simultaneously replaced by alanines yielded crystals suitable for structure determination (Supplementary Table 1, PDB-ID: "7P1W" [https://doi.org/10.2210/pdb7P1W/pdb]"). Superimposition of wildtype (RemA)$_8$ and (RemA-R51A/R53A)$_8$ shows that both structures are identical (r.m.s.d. of 0.3 Å over 600 Cα-atoms) and the positions of R51 and R53 overlap with the varied alanines in the RemA-R51A/R53A structure (Supplementary Figs. 4a, b). This analysis shows that variations of these arginines does neither impact the quaternary nor the tertiary structure of RemA but impairs DNA-binding.

Next, we analyzed the effects of both arginine mutants of (Bs) RemA on the in vivo activity of the epsA promoter (PepsA) using a transcriptional PepsA-lacZ reporter fusion as the readout, as shown previously[33]. The PepsA-lacZ reporter fusion strains carried a deletion of the native remA gene and harbor an integration of the B. subtilis wildtype (Bs)remA gene or its (Bs)remA-R50A and (Bs) remA-R51A derivatives as single copy constructs at the non-essential ('ytnM-ytoI') locus. Transcription of remA was controlled by the IPTG-inducible Phy promoter. To avoid the repression of PepsA-lacZ transcription by SinR[6,8], we genetically interrupted its structural gene in the reporter strains. LacZ activity assays clearly showed activation of PepsA promoter activity in the presence of the wildtype remA gene, whereas the R50A and R51A RemA variants were unable to support PepsA promoter activity (Fig. 2f).

Next, we performed colony biofilm assays to assess the role of arginines 50 and 51 of (Bs)RemA. While the B. subtilis wildtype strain DK1042 formed structured colonies on solid surfaces, the RemA disruption strain (ΔremA) was unable to form biofilms (Fig. 2g). As expected, complementation of the ΔremA strain by a chromosomally integrated and IPTG-inducible copy of the (Bs) remA gene rescued biofilm formation (Fig. 2g). In contrast, the B. subtilis strains carrying copies of the (Bs)remA-R50A and -R51A variants were unable to form structured biofilms (Fig. 2g). Taken together, these findings show that DNA-binding is achieved through the outer surface of the 8-mer of RemA, specifically involving the conserved arginines 50 and 51 (Supplementary Fig. 5).

**(RemA)$_8$ can form a functionally relevant 'back-to-back' 16-mer.** Purified (Gt)RemA showed two peaks when analyzed by SEC (Fig. 1b, Supplementary Figs. 1a, b). Our SEC-MALS analysis indicated that peak-1 exhibits a molecular weight equivalent to a 16-mer of RemA, which could be viewed as a dimer of (RemA)$_8$. To understand this observation in greater details, we re-inspected the crystal packing of the (Gt)RemA structure (Supplementary Fig. 3a). Two (RemA)$_8$ rings are always stacked 'back-to-back' to form 16-mers (Fig. 3a). This association between two (RemA)$_8$ rings is established by the helices α1 in each of the RemA subunits. Closer inspection of the contacts between the α1 originating from two (RemA)$_8$ within the 16-mer shows that their contacts are primarily mediated through hydrogen bonds/salt bridges by the arginines 32, aspartate 36 and glutamate 39 (Fig. 3b). We would like to note that glutamate 39 in (Gt)RemA is a functionally preserved aspartate residue in the B. subtilis protein, whilst all other residues of helix α1 are identical (Fig. 1a). Taken together, our structural analysis suggests that the helices α1 of one (RemA)$_8$ enable the back-to-back interaction with another (RemA)$_8$ ring to establish the formation of a 16-mer of RemA, which possesses a double-donut shape (Fig. 3a).

Support of our structural findings with respect to the wildtype RemA protein was provided by the analysis of a previously isolated loss-of-function missense mutation in (Bs)RemA. In this (Bs)RemA variant (i.e., sor31 for suppressor of sinR;[32]), proline 29 is replaced by a serine. Our structure of (Gt)RemA now shows that proline 29 marks the beginning of helix α1 and appears to be involved in maintaining the relative orientation and structural integrity of every α1 helix within the RemA oligomer (Fig. 3a, d). Indeed, a P29S variant of (Bs)RemA (equivalent to sor31;[32]) was not able to activate PepsA-dependent transcription and also did not allow biofilm formation (Fig. 3c, d).

We also analyzed the behavior of several amino acid substitutions (i.e., R32A, D36S, and D39K/D39A), all of which are positioned in helix α1, in PepsA-lacZ reporter and biofilm assays (Fig. 3c, d). The charged side chain of each of these residues is proposed to provide an interaction interface between the two RemA octamers (Fig. 3b). The D36S and D39A single mutants showed levels in epsA promoter activity comparable with the wildtype RemA protein and allowed the formation of structured biofilms in B. subtilis. However, simultaneous variation of the aspartates 36 and 39 into alanines led to a significantly decreased PepsA-lacZ reporter activity and completely impaired B. subtilis biofilm formation (Fig. 3c, d). Furthermore, alanine substitution of arginine 32 completely abolished PepsA promoter activity and biofilm formation (Fig. 3c, d). Biochemical analysis of the (Gt)RemA variants P29S, R32E, and R32A showed much lower levels of 16-mer formation in ratio to the 8-mer compared to wildtype (Supplementary Fig. 1c). Thus, our experiments indicate that residues of helix α1, which are involved in the back-to-back dimerization of two (RemA)$_8$, are also critical for the regulatory activity of RemA and for B. subtilis biofilm formation.

**Importance of the RemA oligomer architecture and relative subunit geometry.** Previously, Winkelmann et al. reported another loss-of-function missense mutation of B. subtilis remA, named sor4, in which arginine 18 was changed into tryptophan (i.e., R18W[32]). A B. subtilis strain expressing this RemA-R18W variant was unable to form biofilms (Fig. 3d). In our structure of (Gt)RemA, arginine 18 localizes at the inner surface of (RemA)$_8$ (Fig. 3e). The guanidium moiety of arginine 18 of one RemA subunit forms a salt bridge and a hydrogen bond with the carboxyl moiety of aspartate 59 and the backbone carbonyl of alanine 17, respectively, both being from the adjacent RemA subunit (Fig. 3f). Thus, arginine 18 appears to be a critical determinant for the stabilization of the (RemA)$_8$. However, our inspection of wildtype (Gt)RemA structure did not provide any satisfactory answer explaining the molecular consequences of the R18W replacement in the sor4 remA allele. Thus, we decided to determine the structure of the (Gt)RemA-R18W variant.

During the purification of the protein, we observed - like for the wildtype protein (Fig. 1b) - two peaks during SEC analysis (Supplementary Fig. 1d). However, SEC-MALS analysis of both peaks already showed that the RemA-R18W forms smaller oligomers with molecular weights of 123 kDa ± 11% and 71 kDa ± 8%, likely corresponding to 14- and 7-mers, respectively (Supplementary Fig. 1d). These findings were confirmed by the crystal structure of the (Gt)RemA-R18W variant (PDB-ID: "7BME"), which was determined to a resolution of 2.6 Å by molecular replacement employing the crystal structure of wildtype (Gt)RemA as search model (Supplementary Table 1).

In stark contrast to the wildtype protein, the (Gt)RemA-R18W variant forms donut-shaped 7-mers instead of 8-mers (compare: Fig. 3e, g). This change in oligomerization state is accompanied by a reduction in diameter of the ring (i.e., from 72–63 Å) and a concomitant decrease of the interface area between two RemA-R18W monomers (i.e., from 954–879 Å$^2$). Thus, our structural analysis suggests that a major reason for the change of the 8-mer into a 7-mer is that tryptophan 18 is unable to interact with Asp59 and Ala17 of the neighboring subunit and thereby can no longer support the subunit orientation required for the stable formation of (RemA)$_8$ (Fig. 3f). An important consequence of the formation of (RemA-R18W)$_7$ is that the arginines 50 and 51, which are crucial for DNA-interaction in the wildtype protein (see above), show a different spacing and orientation in the 7-mer (Supplementary Fig. 4c). This feature is likely the reason why the RemA-R18W variant is unable to support the transcription activity of the PepsA promoter and to activate biofilm formation (Fig. 3d, h). The observation that the RemA-R18W variant is still able to shift the PepsA promoter-containing DNA in an in vitro EMSA assay (Supplementary Fig. 4d), but is clearly defective in vivo (Fig. 3d, h) shows that the (RemA)$_8$ is required to address a specific DNA topology to activate its target promoters. Perhaps,

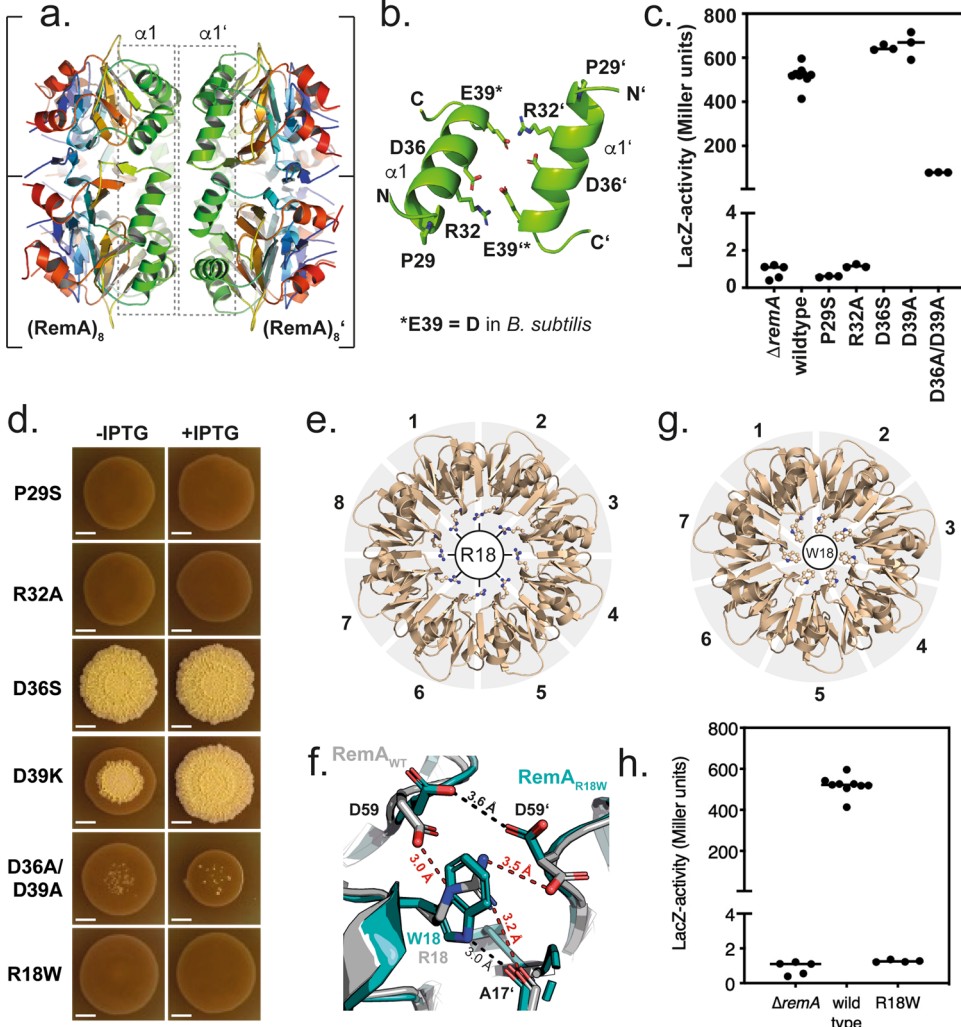

**Fig. 3 Structural and functional architecture of the biofilm regulator RemA. a** Two (RemA)$_8$ align *back-to-back* via their α1-helices to form a 16-mer. The color code is as defined in Fig. (1a). **b** Within the 16-mer of RemA, helices α-1 mediate the interactions through electrostatic interaction. Proline 29 marks the beginning of helix α1. **c** LacZ promoter activity assays of wildtype RemA and its variants carrying amino acid changes within helix α1. Activity assays of a *remA* mutant strain ($n = 5$) and of complementation strains encoding inducible (+ IPTG), ectopic copies of wildtype RemA ($n = 9$), RemA-P29S ($n = 3$), RemA-R32A ($n = 3$), RemA-D36S ($n = 3$), RemA-D36A-D39A ($n = 3$), and RemA-D39A ($n = 3$) are presented. Each point represents the activity measured in a biological replicate. Data are presented as mean from independent experiments. Source data are provided as a source data file. **d** Biofilm formation of different RemA mutant strains. The scale bars shown correspond to 0.5 cm. Biofilm formation was analyzed in at least two independent experiments for each strain with several independently grown colonies per experiment. The colonies shown represent typical examples. **e** Front view of (RemA)$_8$ showing the positions of the arginines 18 of each subunit. **f** Arginine 18 established contacts between two RemA subunits of (RemA)$_8$. Replacement of arginine 18 into tryptophane in the RemA-R18W variant leads to a different contact pattern. **g** Front view of (RemA-R18W)$_7$ showing the positions of the tryptophanes 18 of each subunit. **h** LacZ promoter activity assays of wildtype RemA ($n = 9$) and the RemA-R18W variant ($n = 4$) in the presence of IPTG. Each point represents the activity measured in a biological replicate. Source data are provided as a source data file.

the RemA-R18W variant might also be impaired in its interaction with other proteins required for the biological function of RemA. Taken together, our structural comparison of wildtype and R18W variant of RemA shows that arginine 18 and its intermolecular interactions within (RemA)$_8$ ensures the correct spacing between the RemA subunits. It is therefore an important structural determinant for the overall architecture of the RemA ring. These findings underscore the importance of subunit stoichiometry and geometry of (RemA)$_8$ for its biological function.

## Discussion
The RemA protein is crucial for biofilm formation in *B. subtilis* (Fig. 2g;[32]), yet its role in this cellular differentiation process is widely underappreciated and only poorly understood at the

mechanistic level. Our study now shows that RemA forms an unusual donut-shaped ring structure composed of eight monomers. Moreover, two (RemA)$_8$ rings can dynamically dimerize in a concentration-dependent manner into a 16-mer. This dimerization is mediated via the electrostatic properties of the helices α1, which decorate the backside of each (RemA)$_8$. Moreover, RemA interacts with DNA through its positively charged outer surface via several positively charged arginine residues located on the lateral side of the octameric ring. This raises the question of how DNA-binding by the RemA oligomer possibly looks like from a structural perspective.

In order to gain further insights, we made use of the structural similarity between the LytTR-type DNA-binding domain of the AgrA response regulator from *S. aureus* bound to DNA (AgrA-C/DNA)[37,38], and a homodimeric unit within (RemA)$_8$ (Fig. 2d).

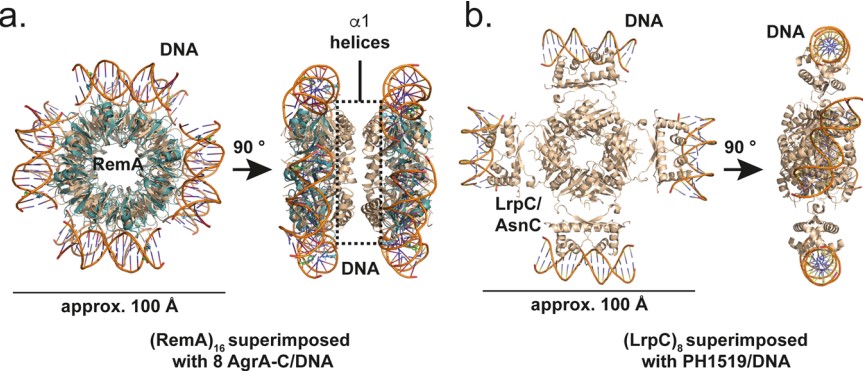

**Fig. 4 Structural comparison of the DNA-binding protein oligomers RemA and LrpC. a** Superimposition of eight DNA-bound AgrA-Cs (cyan; PDB-ID: "3BS1") onto (RemA)₁₆ (orange; PDB-ID: "7BM2"). **b** Structure of the transcriptional regulator LrpC from *B. subtilis* (PDB-ID: "2CFX") superimposed with PH1519 domain bound to DNA (PDB-ID: "2E1C").

Superimposition of the AgrA-C/DNA structure onto the structure of the RemA oligomer resulted in the positioning of four or eight DNA fragments at the outer surface of (RemA)₈ and (RemA)₁₆, respectively (Fig. 4a). From a structural point of view, the RemA oligomer employs multiple DNA-binding sites that likely act in concert. Results published by Winkelman et al. support this hypothesis[33]. These authors showed by in vitro transcription experiments that a DNA fragment with a length of 107–159 base pairs (bp) upstream of the P*epsA* transcription start site is required to fully activate the RemA-dependent transcription of this promoter. DNaseI footprint experiments revealed at least six periodic 7-bp regions of protection within this regulatory region[33]. These AT-rich direct repeats show only a modest level of DNA-sequence conservation and are spaced by 3-bp long intervals of either unprotected or enhanced DNaseI digestion. Winkelmann et al. speculated that bending of the DNA and widening of the major groove of the target DNA during RemA binding might be the reason for this protection pattern[33]. This suggestion fits nicely with the functional importance of the arginines 50 and 51 of RemA for DNA-binding. This finding suggests that the interaction of RemA occurs through the minor groove side of the DNA, because arginines are well known to promote minor groove binding in many DNA-binding proteins (e.g.,[39]).

The possible DNA arrangement and the dimensions of the RemA octameric ring are reminiscent to that observed earlier for transcriptional regulators of the AsnC/LrpC-class in bacteria and archaea (Fig. 4b). These proteins also assemble into octamers and were proposed to bind DNA in a nucleosome-like manner[40,41]. AsnC/LrpC-type proteins consist of an N-terminal helix-turn-helix (HTH) DNA-binding domain and a C-terminal regulatory domain that is often involved in amino acid binding[40]. Despite some similarities at first glance, both protein assembly and DNA-binding is fundamentally different between RemA and AsnC/LrpC. While RemA forms rings, the LrpC assembly is formed by four dimers that expose their HTH domains on the outside to allow for DNA-binding (compare: Fig. 4a, b). RemA can further assemble into 16-mers and although the crystal packing suggests a potential 16-mer of AsnC/LrpC[40], only octamers bound to DNA were observed upon ligand binding[42]. Lastly, AsnC/LrpC proteins bind DNA through classical HTH motifs, while RemA features DNA-binding via flexible regions on the outside of the ring (Fig. 2).

Our structural, biochemical and functional data strongly suggest that not only (RemA)₈ but also the (RemA)₁₆ oligomer is of functional relevance (Fig. 3c, d). Thus, we expect that (RemA)₁₆ would provide a binding surface to wind two adjacent DNA fragments around the (RemA)₁₆ particle (Fig. 4a). If true; one

would have expected 16 rather than the 6 periodic 7-bp regions of protection, which were identified through DNaseI protection assays at the P*epsA* and the P*tapA* promoters[33]. These footprinting experiments were conducted with a (*Bs*)RemA protein fusion carrying a large N-terminal solubility tag (MBP). Our biochemical analysis now shows that solubility tags hinder (RemA)₈ to form (RemA)₁₆ (Supplementary Fig. 2). Due to these technical limitations, the maximum number of DNaseI footprints that a native RemA could leave at its target promoters can currently not be addressed by straightforward experiments. Thus, future investigations need to solve the solubility problem to provide further biochemical and structural insights into the RemA/DNA super complex. This information is particularly important to understand the interplay of the biofilm activator (RemA)₈, and eventually (RemA)₁₆, with the biofilm repressor SinR at the P*epsA* and the P*tapA* promoters[27,28]. It is clear that the binding sites of the SinR repressor and that of the RemA activator overlap in the *epsA-O* and *tapA-sipW-tasA* regulatory regions[33]. SinR is a tetramer with an N-terminal DNA-binding domain in each of its subunits, while the C-terminal domains mediate tetramerization[23]. Consequently, DNA-looping occurs both at the P*eps* and P*tap* promoters once SinR is bound. As a next step, it is important to study how RemA and its oligomerization properties can relieve the SinR-mediated repression to promote transcription of genes central for biofilm formation.

Since RemA activity is crucial for the activation of biofilm-promoting operons even in the absence of SinR[32,33], a SinR independent mechanism of transcriptional activation appears to exist. RemA binds and wraps DNA regions far upstream of the *eps* or *tap* promoter core elements. Transcription factor-dependent wrapping, bending or distortion of the DNA duplex in these upstream regions have been shown to be regulatory principles that modulate compaction of the downstream DNA and thereby affect RNA polymerase (RNAP) binding and transcriptional activity (e.g., H-NS, Fis, CAP; summarized in:[43,44]). This let us speculate that a similar mode of transcriptional activation is also true for RemA activity and that wrapping of upstream DNA around (RemA)₈ or (RemA)₁₆ enhances the formation of the closed RNAP-DNA complex and subsequent transcription. Hence, SinR function can be interpreted as anti-activation mechanism[33]. As a next step, it is important to study how RemA and its oligomerization properties and the SinR repressor antagonize each other to modulate the transcription activity of genes central for biofilm formation.

Our structural analysis showed that a dimer positioned within (RemA)₈ is very similar to a monomer of the C-terminal, DNA-binding domain of the response regulator AgrA (AgrA-C)[37,38].

Full-length AgrA consists of an N-terminal CheY-like receiver domain, which is attached via a flexible linker to AgrA-C (Supplementary Fig. 3b). AgrA-C comprises a LytTR DNA-binding domain, which is found in regulatory proteins involved in the virulence regulation of many bacterial pathogens[38,45,46]. AgrA-C exhibits a two-fold symmetry among its 10 β-strands with loop regions interacting with two consecutive major grooves at their target DNA[37,38,47]. Already in the first structural characterization of AgrA-C, Sidote et al. noticed that the evolutionary formation of the LytTR-fold of AgrA may have been derived from the duplication of a minimal unit[37]. Our study now shows that such a minimal unit is found in the RemA monomer, although the evolutionary relationship between RemA and AgrA-C is unclear. Interestingly, four AgrA-C molecules form a ring-like structure in the crystal lattice that are reminiscent of RemA rings (Supplementary Fig. 3d). This crystallographic artefact suggests that an isolated AgrA-C has kept the general ability to form rings exposing the DNA-binding loops (Supplementary Fig. 3e). However, the full-length AgrA is primarily a monomer in the solution that can dimerize in a phosphorylation-dependent manner to subsequently interact with target DNA to activate transcription.

Our structural analysis of a previously identified suppressor variant in (Bs)remA, RemA-R18W[32], shows that one amino acid exchange is sufficient in order to change the quaternary structure of this protein to form 7- and 14-mers instead of 8- and 16-mers formed by the wildtype protein, respectively. Since we showed that RemA-R18W can bind DNA (Supplementary Fig. 4d), one wonders if evolution could make use of this novel oligomeric assembly to control gene expression eventually in a new physiological context.

Taken together, our study shows that RemA is an unusual bacterial DNA-binding protein, which recognizes its target DNA partner through multiple sites. Importantly, the environmental and/or cellular cues that dictate the DNA-binding activity of RemA remain to be elucidated. The regulatory function of RemA is not confined to biofilm formation by *B. subtilis*, as systems for cellular adjustment to osmotic stress and nitrogen metabolisms are also part of the RemA regulon[33]. Notably, orthologues of RemA are present not only in Firmicutes to which *B. subtilis* belongs, but also in Thermotogales, Cyanobacteria, delta-Proteobacteria, and Chloroflexi[33]. Studies of these RemA-like proteins should be aided substantially by the structural and functional analysis presented here.

## Methods

**Plasmid construction and mutagenesis**. All plasmids and primers used in the study are summarized in Supplemental Table 2. For all cloning experiments, the *E. coli* strain TOP10 (Invitrogen, Carlsbad, CA, USA) was used. *ytnM-ytoI*::P_hy-*remA*. To characterize (Bs)RemA and its variants in vivo, we placed the wild type and mutated (Bs)*remA* genes under the control of the IPTG-inducible P_hy promoter (P_hy-*remA*), respectively. For this purpose, we amplified the *remA* gene, including its native ribosome binding site, from the chromosome of *B. subtilis* JH642 using the primers remA-RBS-for and remA-rev (Supplemental Table 2). The resulting DNA fragment was cleaved with *Hind*III and *Nhe*I and inserted into plasmid pDR111 downstream of the P_hy promoter (D. Rudner, Boston, MA, US). This yielded plasmid pTMB33. To allow integration of the (P_hy-*remA*) construct into the chromosome of *B. subtilis* at the non-essential intergenic *ytnM-ytoI* region we used plasmid pBB284 (D. Rudner, Boston, MA, US) that carries a 5′-flanking region covering part of the *ytnM* gene and a 3′-flanking region covering part of the *ytoI* gene. We sub-cloned the *EcoRI-BamHI* fragment from pTMB33 (*lacI-*P_hy-*remA*) between the 5′- and 3′ flanks into the *EcoRI-BamHI* opened pBB284 backbone thereby yielding plasmid pTMB42. Codons in *remA* were exchanged by site-directed mutagenesis. For this purpose, custom-synthesized primers with corresponding nucleotide exchanges (Supplemental Table 2) were used to amplify plasmid pTMB42 with the Q5-Site-Directed Mutagenesis Kit (New England Biolabs, Frankfurt, Germany). The mutated *remA* sequences were verified by DNA sequencing (Microsynth, Balgach, Switzerland). *amyE*::P_epsA-*lacZ*. To generate the P_eps transcriptional reporter fusion construct to β-galactosidase (*lacZ*), a PCR product containing the P_eps promoter was amplified from *B. subtilis* 3610 chromosomal DNA using the primers (7539/3025, Supplemental Table 2). The PCR product was digested with *EcoRI* and *BamHI* and cloned into the *EcoRI* and

*BamHI* sites of plasmid pDG268[48], which carries a chloramphenicol-resistance marker and a polylinker upstream of the *lacZ* gene between two arms of the *amyE* gene to generate the plasmid pFC1.

**Strain construction**. See Supplementary Table 3 for an overview of all strains and their construction. To analyze the impact of various (Bs)RemA variants in vivo, we constructed *B. subtilis* strains that carry chromosomal integrations of the wild type and corresponding mutated (Bs)*remA* genes under the control of the IPTG-inducible P_hy promoter (P_hy-*remA*). We used DNA of plasmid pTMB42, or its mutagenized derivatives, cleaved them with *PvuI* and transformed the marker exchange strain TMB410 (JH642 ´*ytnM-ytoI*::*cml*R) with the linearized plasmids. Transformants were selected on LB agar plates containing spectinomycin and were subsequently screened for chloramphenicol sensitivity as an indication for integration of the (*lacI-*P_hy-*remA*, *spc*R) construct at the *ytnM-ytoI* intergenic region via a double homologous recombination event. We verified correct chromosomal integration and the sequence of the wild type or mutant *remA*-genes by PCR and DNA sequencing (Microsynth, Balgach, Switzerland). P_epsA-reporter strains: To quantify the influence of (Bs)RemA and its mutant variations on *epsA* promoter activity, we constructed a chassis strain (TMB524) that carried a *remA*::*zeo*R insertion-deletion allele and a *sinR*::*kan*R insertion-deletion allele to fully de-repress *eps* transcription. *sinR*::*kan*R: The *sinR*::*kan*R insertion-deletion allele was generated by long flanking homology PCR[49] (LFR) using primers 403 and 404, 405, and 406, and DNA containing a kanamycin drug resistance gene (pDG780) was used as a template for marker replacement[50]. The resulting product was used to transform PY79 selecting for kanamycin resistance and transduced into *B. subtilis* strain 3610 using SPP1-mediated generalized transduction thereby yielding strain DS859. *remA*::*zeo*R: The *remA*::*zeo*R insertion-deletion allele was generated by long flanking homology fusion PCR[49]. *remA* flanking regions were amplified from chromosomal DNA of *B. subtilis* JH642 using primers remA-P1-for, remA-P2-zeo(anti)-rev, and remA-P3-zeo(anti)-rev and remA-P4-rev, and a zeocin resistance gene was amplified with plasmid p7Z6[51] as DNA template (primers remA-RC-P2-zeo(anti) and remA-RC-P3-zeo(anti)). Finally, the P_epsA-*lacZ* fusion allele was transduced from strain FC5 into strain TMB523 thereby yielding the final chassis strain TMB524. We then inserted the (*lacI-*P_hy-*remA*, *spc*R) constructs of the (Bs)*remA*-wild type-gene or corresponding mutated versions of (Bs)*remA* as single copies into the *ytnM-ytoI* intergenic region of TMB524 via SPP1-mediated transduction (Supplementary Table 3). Biofilm strains: To analyze the impact of various (Bs)RemA variants on biofilm formation we used the *remA*::*tet*R strain DK7212, a derivative of the non-domesticated *B. subtilis* NCIB3610 wildtype strain. The *remA*::*tet*R insertion-deletion allele present in the *B. subtilis* strain DK7212 was generated by long flanking homology PCR (using primers 1087 and 1088, 1089, and 1090), and DNA containing a tetracycline drug resistance gene (pDG1515) was used as a template for marker replacement[50]. The resulting product was used to transform *B. subtilis* strain DK1042[52] selecting for tetracycline resistance. Linearized DNA of plasmid pTMB42, or its mutagenized derivatives, was then used to transform DK7212 selecting for spectinomycin resistance. Correct integration of the (*lacI-*P_hy-*remA*, *spc*R) construct at the *ytnM-ytoI* intergenic region was verified by PCR and DNA sequencing (Microsynth, Balgach, Switzerland).

**Protein production and purification**. The genes encoding RemA from *B. subtilis* strain 3610 and *G. thermodenitrificans* NG-80 were amplified from the respective genomic DNA and cloned into the pET24d vector (Novagen) via the *NcoI* and *XhoI* restriction sites with an N-terminal hexa-histidine tag (primers and plasmids are summarized in Supplementary Table 4). RemA protein was produced in BL21 (DE3) (Novagen). For gene expression, *E. coli* BL21(DE3) was grown in LB medium under autoinduction conditions [D ( + )-lactose-monohydrate, 0.5% (w/v)] supplemented with the respective antibiotic (50 µg ml$^{-1}$ kanamycin) at 30 °C for 16 h under constant shaking (180 rpm). After centrifugation, cells were resuspended in buffer A (20 mM HEPES, 20 mM MgCl$_2$, 20 mM KCl, 1000 mM NaCl, 40 mM imidazole, pH 8). After cell lysis by a Microfluidizer (M110-L, Microfluidics), cell debris was removed by high-speed centrifugation. The clarified lysate was applied to a 5 mL FF HisTrap column (GE Healthcare), immobilized (Gt)RemA washed with buffer A and eluted with buffer B (like buffer A but supplemented with 500 mM imidazole). All (Gt)RemA protein and its variants used in this study were applied to size exclusion chromatography (16/60 S200 Superdex, GE Healthcare), equilibrated in buffer A without imidazole. Fractions containing (Gt)RemA as verified by Coomassie-stained SDS-PAGE were pooled and concentrated. Seleno-methionine-derivatized protein was produced as described previously (e.g.,[53]). Briefly, (Gt)RemA with the Se-Met label was expressed in *E. coli* BL21 (DE) cells using M9 medium supplemented with 125 mg lysine, 125 mg threonine, 125 mg phenylalanine, 50 mg valine, 50 mg leucine, 50 mg isoleucine, 5 g glucose, 250 mM MgCl$_2$, 1 mM CaCl$_2$ and 50 mg Seleno-L-methionine per liter. Protein purification was carried out as described for the native proteins.

The His-GB1-fusion variants of (Gt) and (Bs)RemA were produced as described above (primers and plasmids are summarized in Supplementary Table 4). Protein production was performed in an auto inductive lysogeny broth medium containing 1% (w/v) of lactose at 30 °C under constant shaking for 16 h. After cell harvest and lysis, cellular debris was removed by high-speed centrifugation. The His-GB1-RemA fusion variants were enriched by nickel-ion affinity purification at RT (FF-HisTrap

columns; GE Healthcare). The equilibration/ wash buffer (Buffer A) consisted of 20 mM HEPES-Na, 200 mM NaCl, 20 mM KCl, 20 mM MgCl₂ and 40 mM Imidazole (pH 8.0). The elution buffer (Buffer B) had the same composition but contained 250 mM Imidazole. After nickel-ion affinity purification, the His-GB1-fusion variants of (*Gt*) and (*Bs*)RemA were subjected to TEV protease treatment (0.8 mg TEV protease) after buffer exchange to buffer A using an Amicon Ultra-3K centrifugal filter (Merck Millipore). On-column cleavage of the His-tagged GB1 by immobilized TEV protease using a 1 ml FF-HisTrap column and a peristaltic pump was performed at room temperature for 16 h. Tag-less (*Gt*)RemA was collected in the flowthrough of the second (reverse) Ni-NTA chromatography step, whilst the cleaved His-tagged GB1, remaining TEV protease and uncleaved His-GB1-(*Gt*) or (*Bs*)RemA remained bound to the column through their His-tags. Tag-free (*Gt*)RemA protein was subjected to SEC using a Superdex S200 Increase 10/300 column equilibrated in SEC buffer (described above).

**Crystallization and structure determination**. Crystallization was performed by the sitting-drop method at 20 °C in 0.5 µl drops consisting of protein and precipitation solutions in ratios of 1:1 and 1:2. (*Gt*)RemA crystallized at 40 mg ml⁻¹ concentration within 1 day in 1.6 M ammonium sulfate, 0.1 M citric acid pH 3.5, final pH 4. (*Gt*)RemA-R18W crystallized at 30 mg ml⁻¹ within 2 days in 2.5 M NaCl, 0.2 M Li₂SO₄, 0.1 M NaOAc pH 4.5. (*Gt*)RemA-R51A/R53A crystallized at 30 mg ml⁻¹ within 2 days in 0.1 M Tris pH 8.5 and 2 M NH₄H₂PO₄. Prior to data collection, crystals were flash-frozen in liquid nitrogen employing a cryo-solution that consisted of mother-liquor supplemented with 30% (v/v) glycerol. Data were collected under cryogenic conditions at the European Synchrotron Radiation Facility (Grenoble, France) at beamlines ID23-1 and ID30A-1. Data were processed with XDS[54] and ccp4-implemented AIMLESS[55]. The (*Gt*)RemA structure was determined by experimental phasing using the selenium single anomalous dispersion (Se-SAD). The substructure was determined with Crank-2[56], manually built in Coot[57], and refined with PHENIX[58]. The structures of (*Gt*)RemA-R51A/R53A and (*Gt*)RemA-R18W were solved by Molecular Replacement in Phenix using a monomer of (*Gt*)RemA (this study) as a search mode[58]. Figures of protein structures were prepared with Pymol[59] and UCSF Chimera[60].

**Media and growth conditions**. *E. coli* and *B. subtilis* strains were grown on Lysogeny broth (LB) agar plates or LB liquid medium for plasmid or strain constructions. When appropriate, antibiotics were added at the following concentrations: chloramphenicol (10 µg ml⁻¹), kanamycin (5 µg ml⁻¹), spectinomycin (100 µg ml⁻¹), tetracycline (10 µg ml⁻¹), zeocin (50 µg ml⁻¹) and ampicillin (100 µg ml⁻¹). A final concentration of 1 mM IPTG was used to induce the *Phy* promoter. For LacZ reporter experiments, cultures were grown in 20 ml LB medium (in a 100 ml Erlenmeyer flask), at 37 °C. Biofilm colonies of the un-domesticated *B. subtilis* strain NCIB3610 were grown on MSgg agar plates containing 5 mM potassium phosphate (pH 7), 100 mM MOPS (pH 7), 2 mM MgCl₂, 700 µM CaCl₂, 50 µM MnCl₂, 50 µM FeCl₃, 1 µM ZnCl₂, 2 µM thiamine, 0.5% (v/v) glycerol, 0.5% (w/v) glutamate and 1.5% (w/v) agar[9]. MSgg plates were inoculated with 10 µl of an LB overnight culture. Biofilms were allowed to grow for four days at 30 °C. At least two biofilm macro-colonies were grown from different cultures in two independent experiments conducted on different days. Agar plates were positioned onto a black background with one-side illumination. Macro-colonies were then imaged with a digital reflex camera D5600 (Nikon, Düsseldorf, Germany) in the automatic mode (mode P) for exposure time. The camera ISO was set to 650.

**SPP1 phage transduction**. SPP1-mediated generalized phage transduction was used to transfer chromosomal gene alleles from domesticated *B. subtilis* JH642 derivatives into non-domesticated *B. subtilis* NCIB3610 backgrounds. The preparation of lysates and the transduction followed a procedure described previously[32]. In short, lysates were prepared by mixing bacterial culture with serial diluted SPP1-phage stocks, incubating for 15 min at 37 °C, adding molten TY soft agar (0.3%) and spreading on TY agar (1.5%) plates. After incubation for 16 h at 37 °C plates were analyzed for plaque formation. Phages were harvested by addition of TY medium, scraping of the top agar layer, following sedimentation, and the passage of the supernatant through a 0.2 µm-pore-size syringe filter.

Cells of stationary *B. subtilis* cultures were transduced with the phage SPP1 by addition of 30 µl of the phage lysate (see above) to 1 ml of recipient cells. TY broth was subsequently added to the mixture, followed by incubation at RT vigorously shaking for 30 min. The transduction mixture was then centrifuged, the supernatant discarded, and the pellet was resuspended in the remaining volume. The cell suspension was then plated onto LB agar including the appropriate antibiotics to select for the transduced marker genes, and 10 mM sodium citrate to reduce SPP1-phage reinfection.

**β-Galactosidase assay**. We grew the reporter strains in LB medium and harvested 1.5 ml of the cultures when they reached an optical density (OD) at a wavelength of 578 nm of 2. Resuspension of the pellets, cell lysis, and determination of β-galactosidase activity was performed as described previously[32]. Cells were suspended in 1.5 ml of Buffer Z (40 mM NaH₂PO₄, 60 mM Na₂HPO₄, 1 mM MgSO₄, 10 mM KCl, 38 mM 2-mercaptoethanol and after addition of lysozyme (0.2 mg ml⁻¹), each sample was incubated at 30 °C for 15 min. Subsequently, 500 µl of the cell lysates were

used for the enzyme assay. If required, the cell lysates were diluted with buffer Z. The β-galactosidase enzyme reaction was started by the addition of 100 µl of a solution of 4 mg of 2-nitrophenyl β-d-galactopyranoside ml⁻¹ and stopped by the addition of 250 µl of a 1 M Na₂CO₃ solution. The OD₄₂₀ of the reaction mixture was measured, and the β-galactosidase-specific activity was calculated as follows: [OD₄₂₀/(time × OD₆₀₀)] × dilution factor × 1000. Prism 9 software (GraphPad, San Diego, CA, US, version 9.0.2 for macOS) was used for the mean calculation of data from at least three independent experiments and for the creation of scatter plots.

**Electrophoretic mobility shift assay (EMSA)**. EMSAs were carried out to analyze the DNA-binding activity of wildtype (*Gt*)RemA and its variants. A fluorescently labeled 289 bp PCR fragment covering the P*epsA* promoter region was amplified from *B. subtilis* JH642 chromosomal DNA. In a binding reaction, 1 pmol of the DNA fragment was mixed with the indicated protein concentrations in EMSA buffer containing 20 mM HEPES (pH 8.0), 1000 mM NaCl, 20 mM MgCl₂, 20 mM KCl, 0.3 µg ml⁻¹ bovine serum albumin, 25 µg ml⁻¹ herring sperm DNA, 10% (v/v) glycerol and 0.25 mg ml⁻¹ Orange-G dye in a final volume of 14.5 µl. After incubation of the reaction mixture at 37 °C for 15 min, samples were loaded onto a native 12% (w/v) polyacrylamide gel (in 1 × TTE containing 90 mM Tris Base, 30 mM Taurine, and 1 mM EDTA). Samples were separated at 150 V for 90 min and subsequently imaged with the 800 nm channel of an Odyssey FC Imager (LI-COR Biosciences, Lincoln, US).

**Reporting summary**. Further information on research design is available in the Nature Research Reporting Summary linked to this article.

## Data availability
The coordinates and structure factors generated in this study have been deposited in the Protein Data Bank (PDB) under the accession codes "7BM2", "7BME", and "7P1W" for wildtype RemA, RemA-R18W, and RemA-R51A/R53A, respectively. Plasmids, primers, and strains associated with this manuscript are available upon request to either G.B. or E.B. The LacZ activity, size exclusion chromatography (SEC), multi-angle light scattering (MALS), and EMSA data generated in this study are provided in the Source Data file. Source data are provided with this paper.

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

## Acknowledgements

G.B. and E.B. thank the Deutsche Forschungsgemeinschaft (DFG) for financial support through the Collaborative Research Council CRC987 project. We acknowledge Dr. A. Lepak for his technical input into the project and are very grateful to J. Gade for her excellent technical support during strain constructions and physiological experiments. We thank D. Rudner (Harvard Medical School, Boston, USA) for providing expression plasmids. We are grateful to Felix Dempwolff and Kursad Turgay for critically reading the manuscript.

## Author contributions

T.H., D.M. and P.B. conceived and performed experiments. F.B. and C.N.M. performed SEC analysis. D.M. and F.A. determined crystal structures. D.B.K. provided strains. E.B., T.H., P.B. and G.B. wrote the paper. All authors discussed the results and commented on the paper.

## Funding

## Competing interests

The authors declare no competing interests.
