## [Peer Review File · Nature Communications]

Structural and functional characterization of the bacterial biofilm activator RemAEditorial Note: This manuscript has been previously reviewed at another journal that is not operating a transparent peer review scheme. This document only contains reviewer comments and rebuttal letters for versions considered at *Nature Communications*.

REVIEWERS' COMMENTS

Reviewer #2 (Remarks to the Author):

Hoffman et al. present a novel crystal structure of an important biofilm regulator, RemA, from the model organism *Bacillus subtilis*. They provide evidence that RemA forms a multimeric structure in vitro and that this is the functional form of RemA for transcriptional regulation. The authors combine structural analysis, in vitro DNA binding, and in vivo biofilm assays to show that RemA binds DNA in a novel manner with similarity to the LytTR-family. This is a well written and informative paper that we believe would be well received by the structural biology and microbiology communities.

The paper references relevant literature. The main conclusions are well founded and more clearly explained than the previous iteration.

The authors have addressed most of our points of concern from our initial review. A few minor suggestions/improvements are listed below.

Suggestions:

Methods – Perhaps I missed it, but I still see no reference to how the biofilms were imaged. What scope/camera etc.

Line 61 – Should read "...are derepressed..." as the sentence is talking about the regulation of multiple genes.

Line 233 – "Thus, our experiments conclusively indicate show that..." would be better as "Thus, our experiments indicate that.." Not only is there a redundancy in the sentence but the statement is too strong. Although the results are supportive of your hypothesis, there are potentially other reasons for this (ie. the $\alpha 1$ helix is involved with interaction with other proteins that are necessary for instance).

Line 257 – Would read better "In stark contrast to the wildtype protein, the (Gt)RemA-R18W variant does form donut-shaped..."

Line 268-272 – This conclusion is one possibility. Is it also possible that other interactions might be affected? Perhaps RemA doesn't interact with other proteins and the transcription machinery isn't properly recruited.

Line 300 – typo with a rouge bracket at the end of the sentence

Figure 2 – g – We presume the columns have +/- IPTG associated with them? Please indicate the addition of IPTG where needed in all figures – e.g. lacZ analyses .

Figure 3 – Labels on the x-axis of the graph in part C are a little hard to read. The D36A/D39A would be better as a single line.

Supplementary table 1. – The Rwork and Rfree values are reported as percentages for the first two columns and not for the third. Please correct for consistency.

The references need italics adding and some capital letters adding – but perhaps this is done by the editorial office.

Additional check requested by Editor

Point made by reviewer 3 - "It would be useful if the authors could report DNA-binding activity for the R18W mutant in vitro. It may be the case that this mutant can still bind DNA but perhaps has some defect specific for in vivo conditions (such as a concentration-dependent defect)."

The authors have included both in vivo and in vitro analyses of the ability of R18W to bind to DNA and activate transcription. There is a typo in the text. Line 270 should refer to Fig S4d not Fig S3d.

I am otherwise happy with the changes made to the text.

Reviewer #2 (Remarks to the Author):

Hoffman et al. present a novel crystal structure of an important biofilm regulator, RemA, from the model organism *Bacillus subtilis*. They provide evidence that RemA forms a multimeric structure in vitro and that this is the functional form of RemA for transcriptional regulation. The authors combine structural analysis, in vitro DNA binding, and in vivo biofilm assays to show that RemA binds DNA in a novel manner with similarity to the LytTR-family. This is a well written and informative paper that we believe would be well received by the structural biology and microbiology communities.

Thank you very much.

The paper references relevant literature. The main conclusions are well founded and more clearly explained than the previous iteration.

The authors have addressed most of our points of concern from our initial review. A few minor suggestions/improvements are listed below.

We thank the reviewer and the editor for the positive evaluation of our revised manuscript and the helpful suggestions.

Suggestions:

Methods – Perhaps I missed it, but I still see know reference to how the biofilms were imaged. What scope/camera etc.

We inserted now additional information for this section (line 521) into the manuscript.

Line 61 – Should read “...are derepressed...” as the sentence is talking about the regulation of multiple genes.

Thank you, changed as suggested.

Line 233 – “Thus, our experiments conclusively indicate show that...” would be better as “Thus, our experiments indicate that..” Not only is there a redundancy in the sentence but the statement is too strong. Although the results are supportive of your hypothesis, there are potentially other reasons for this (ie. the $\alpha 1$ helix is involved with interaction with other proteins that are necessary for instance).

Thank you very much for pointing this out. Changed as suggested.

Line 257 – Would read better “In stark contrast to the wildtype protein, the (Gt)RemA-R18W variant does forms donut-shaped...”

We changed it as suggested to: “..., the (Gt)RemA-R18W variant forms donut-shaped ...”

Line 268-272 – This conclusion is one possibility. Is it also possible that other interactions might be affected? Perhaps RemA doesn't interact with other proteins and the transcription machinery isn't properly recruited.

True. That might be. We updated the manuscript accordingly by stating: “...is required to address a specific DNA topology to activate its target promoters. Moreover, the RemA-R18W variant might also be impaired in its interaction with other proteins required for the biological function of RemA. ...”

Line 300 – typo with a rouge bracket at the end of the sentence

Thank you.

Figure 2 – g – We presume the columns have +/- IPTG associated with them?

Has been added. Thank you for pointing this out.

Please indicate the addition of IPTG where needed in all figures – e.g. lacZ analyses .

We added this information into the figure legend, to prevent an overcrowding of the figure.

Figure 3 – Labels on the x-axis of the graph in part C are a little hard to read. The D36A/D39A would be better as a single line.

Has been changed as suggested.

Supplementary table 1. – The R_{work} and R_{free} values are reported as percentages for the first two columns and not for the third. Please correct for consistency.

Thank you. Done.

The references need italics adding and some capital letters adding – but perhaps this is done by the editorial office.

Thank you for pointing this out. We have updated the reference list according to the Nature style.

Additional check requested by Editor

Point made by reviewer 3 - “It would be useful if the authors could report DNA-binding activity for the R18W mutant in vitro. It may be the case that this mutant can still bind DNA but perhaps has some defect specific for in vivo conditions (such as a concentration-dependent defect).”

The authors have included both in vivo and in vitro analyses of the ability of R18W to bind to DNA and activate transcription. There is a typo in the text. Line 270 should refer to Fig S4d not Fig S3d. I am otherwise happy with the changes made to the text.

Thank you for pointing this out. Has been changed.